# Associative Overdominance and Negative Epistasis Shape Genome-Wide Ancestry Landscape in Supplemented Fish Populations

**DOI:** 10.3390/genes12040524

**Published:** 2021-04-03

**Authors:** Maeva Leitwein, Hugo Cayuela, Louis Bernatchez

**Affiliations:** Institut de Biologie Intégrative et des Systèmes (IBIS), Université Laval, Québec, QC G1V 0A6, Canada; hugo.cayuela51@gmail.com (H.C.); Louis.Bernatchez@bio.ulaval.ca (L.B.)

**Keywords:** introgression, associative-overdominance (AOD), stocking, genomic landscape, evolutionary mechanisms, salmonid

## Abstract

The interplay between recombination rate, genetic drift and selection modulates variation in genome-wide ancestry. Understanding the selective processes at play is of prime importance toward predicting potential beneficial or negative effects of supplementation with domestic strains (i.e., human-introduced strains). In a system of lacustrine populations supplemented with a single domestic strain, we documented how population genetic diversity and stocking intensity produced lake-specific patterns of domestic ancestry by taking the species’ local recombination rate into consideration. We used 552 Brook Charr (*Salvelinus fontinalis*) from 22 small lacustrine populations, genotyped at ~32,400 mapped SNPs. We observed highly variable patterns of domestic ancestry between each of the 22 populations without any consistency in introgression patterns of the domestic ancestry. Our results suggest that such lake-specific ancestry patterns were mainly due to variable associative overdominance (AOD) effects among populations (i.e., potential positive effects due to the masking of possible deleterious alleles in low recombining regions). Signatures of AOD effects were also emphasized by highly variable patterns of genetic diversity among and within lakes, potentially driven by predominant genetic drift in those small isolated populations. Local negative effects such as negative epistasis (i.e., potential genetic incompatibilities between the native and the introduced population) potentially reflecting precursory signs of outbreeding depression were also observed at a chromosomal scale. Consequently, in order to improve conservation practices and management strategies, it became necessary to assess the consequences of supplementation at the population level by taking into account both genetic diversity and stocking intensity when available.

## 1. Introduction

Genetic admixture resulting from introgressive hybridization is a fundamental mechanism influencing populations and species evolution [1,2,3,4,5]. This may result from natural secondary contacts following an allopatric period of geographic isolation [3,6]. Genetic admixture may also result from human activities, including biological invasions, human-mediated translocation [7,8,9,10] or the supplementation of wild populations with non-local sources [11,12,13]. Although most of the introgressed genetic variation is likely neutral, admixture may introduce beneficial genetic variants improving the fitness of individuals in the recipient population [5,14,15]. Conversely, introgressed alleles may be maladaptive and have deleterious effects on fitness, for instance due to genomic incompatibilities [16,17,18,19]. Admixture from non-local populations may also have a dilution effect on locally adapted populations [20]. Despite the plethora of studies addressing these issues, few have documented how the interplay between the introduction pressure of foreign alleles (e.g., supplementation history), genetic drift and selection may produce complex admixture landscapes along the genome [21,22].

One important factor to consider for understanding the evolutionary outcomes of admixture is the local variation in recombination rate [13]. In the first generations following hybridization or in low recombining genomic regions, genetic drift and selection (either positive or negative) will act on large genomic tracts [13,22,23]. Hence, two main outcomes are expected: first, if the foreign population has a lower effective size ($N_{e}$) and a stronger genetic load (i.e., accumulation of recessive deleterious alleles) than the recipient population, outbreeding depression may occur due to genetic incompatibilities. Consequently, in the case of negative epistasis (i.e., genetic incompatibilities between the recipient and the foreign population), blocks of introgressed foreign ancestry are expected to be purged more quickly in low recombining regions of the genome [24]. Alternatively, if the genetic diversity of the foreign population is higher than that of the recipient population [11], positive effects may occur in first hybrids generation by masking the effect of link recessive deleterious alleles present in the recipient population (i.e., associative overdominance (AOD), [25,26]). Over time, introgressed tracts will shorten more quickly in highly recombining genomic regions [27]. Therefore, both beneficial or negative selective outcomes are expected to occur following genetic admixture [13,23]. The interplay between variation in recombination rate (which modulates the size of the introgressed tracts) and local genetic diversity is also expected to result in a complex genome wide admixture landscape.

To date, the understanding of such processes shaping genome-wide admixture landscapes remain obscure due to limited number of empirical studies that specifically addressed this issue [13,23]. Consequently, predicting the evolutionary outcomes of biological invasions or the supplementation practices (e.g., genetic rescue, stocking practices) remains challenging and important to develop viable conservation practices. In this study, we empirically investigate the evolutionary mechanisms shaping the genome-wide admixture landscape in fish populations supplemented with a foreign hatchery strain. The Brook Charr (*S. fontinalis*) is a socio-economically important fish species supporting a major recreational fishing industry in eastern North America. Consequently, the species has a long history of intense supplementation (i.e., stocking to supply to angling demand). In Eastern Canada, the domestication of local strains and the development of hatcheries has increased to supplement rivers and lakes over the last century [28,29,30]. Here and further on, we will used the term “domestic” to refer to the hatchery stock that has been used to supplement the populations we studied. Thus, population supplementation performed over the last 50 years in the Province of Québec has led to introgressive hybridization between stocked domestic strains and wild populations resulting in a complex admixture landscape along the genome [31] with an increase in domestic ancestry associated with the increase in stocking intensity [28,32]. Our research group previously evaluated the relationship between the domestic ancestry and the intensity of stocking practices [28], and we subsequently investigated the role of the local recombination rate in shaping the genome-wide ancestry landscape [31]. We observed that the main mechanism determining the domestic ancestry rate was associative overdominance (AOD), indicating that domestic ancestry was mainly favored by masking the effect of linked recessive deleterious alleles [13,31,33]. However, the influence of the genetic diversity and the intensity of stocking effort on the genome-wide landscape of ancestry remained to be investigated. Additionally, the relative contribution of AOD and negative epistasis on the ancestry landscape of the 22 studied populations still remained to be documented at both an intrapopulation level (i.e., among genomes) and an interpopulation level (i.e., similarities between populations). Using an integrative approach, we thus investigated how local population genetic diversity along the genome and the introduction pressure of domestic alleles (i.e., lakes stocking intensity) interplays with AOD and negative epistasis to produce lake-specific genome-wide ancestry patterns.

We predicted three alternative evolutionary scenarios considering the relationships between genome-wide domestic ancestry rate variation and (i) local recombination rate, (ii) local genetic diversity and (iii) lake-specific stocking history (Figure 1A–C). Scenario A corresponds to patterns of associative overdominance, with higher domestic ancestry in low recombining regions and in low-diversity genomic regions [13,31]. We also predicted more domestic ancestry for populations that have experienced higher stocking intensity, given that the probability for hybridization events is expected to increase with the number of fish stocked. Scenarios B and C correspond to negative epistasis predictions, where reduced domestic ancestry is expected in low recombining regions. Scenario B implies the introgression of potential beneficial domestic alleles, whereby we predict locally a lower genetic diversity due to the fixation of the beneficial alleles and a higher introgression rate of domestic ancestry. In Scenario C we predicted lower domestic ancestry in low recombining regions and in genomic regions of lower diversity due to genomic incompatibility. [13,34,35,36]. By taking advantage of a large dataset of 22 supplemented lacustrine populations of Brook Charr in Québec (previously genotyped using Genotype-By-Sequencing (GBS); [28]), we examined which evolutionary scenarios most probably explain the genome-wide domestic ancestry patterns in those populations. We searched for similarities in patterns of ancestry between populations that may reflect adaptive or maladaptive introgression from the domestic strain. Then, we determined which one of the aforementioned alternative scenarios (i.e., AOD vs. negative epistasis) best explained the observed patterns of domestic ancestry across linkage groups.

## 2. Materials and Methods

### 2.1. Study System and Sampling

The dataset used here and produced by Létourneau et al. (2018) comprises the genotypes of 553 Brook Charr collected from 22 isolated lakes from two wildlife reserves (Mastigouche and St-Maurice) in Québec, Canada as well as 37 individuals from the Truite de la Mauricie aquaculture domestic brood stock (Table 1, Appendix A). These lakes have been supplemented with this domestic Brook Charr strain which historically originated from crosses between two populations (Nashua and Baldwin) (details on supplementation history can be found in [3]). The domestic strain has been maintained for more than 100 years, and an average of 6–7 million domestic Brook Charr are released each year into the wild [28,37]. The history of stocking and the domestic strain used for supplementation has been rigorously recorded over time [28,37], and the number of fish stocked per hectare is reported in Table 1. Additional information on the domestic history can be found in Appendix A.

### 2.2. Genomic Data and Local Ancestry Inference

The SNP calling and ancestry inference of the 22 populations have been performed in [4]. Briefly, after applying quality filters, raw reads were demultiplexed with STACKS version 1.40 [38] and aligned to the closely related Arctic charr (S. alpinus) reference genome [39] with BWA_MEM version 0.7.9 [40]. SNP calling was performed separately for each population, and the domestic strain with STACKS version 1.40. Retained RAD loci had a minimum depth of four reads per locus, present in at least 60% of the populations, a minimum allele frequency of 2% and a maximum of 20% of missing data (see further details in Leitwein et al. [4]). To avoid merging paralogs, each individual locus with more than two alleles were removed with the R package STACKR [41]. The local ancestry inference was performed according to Leitwein et al. [5]. RAD loci were mapped against the Artic charr reference genome and ordered along each of the 42 Brook Charr linkage groups (LGs) by applying the MapComp software to the Brook Charr linkage map used as a reference [42]. MapComp allowed retrieving the mapping position of each marker by controlling for synteny and collinearity between the Artic Charr and the Brook Charr genomes [31]. The local ancestry inference was performed with the program ELAI version 1.01 [43] using the domestic strain as the foreign population and wild fish caught in each lake as the admixed recipient populations. ELAI was run 20 times for each 42 linkage groups (LG) to assess convergence. The number of upper clusters (−C) was set to 2 (i.e., assuming that each fish was a mixture of domestic and wild populations), the number of lower clusters (−c) to 15 and the number of expectation-maximization steps (−s) to 20 [31].

### 2.3. Describing Genome-Wide Variation of Domestic Ancestry and Genetic Diversity

We used the ancestry estimates provided in Leitwein et al. [4] to perform subsequent analyses. Briefly, the number and length of domestic tracts within each individual genome was retrieved from ELAI [43] and used to assess hybrids classes. The late-generation hybrids that mostly comprise wild-type ancestry and displayed a chromosomal ancestry imbalance (CAI) of < 0.125 [31,44] were retained for subsequent analyses (Table 1). We chose to remove the early hybrids (e.g., F1, F2, BC1 and BC2) as they were not represented by a sufficient number of fish to perform rigorous statistical analyses. For each population, we then assessed the genome-wide domestic ancestry rate for those late hybrid individuals.

We identified regions presenting excess or deficit of domestic ancestry considering local variation in recombination rate. To do so, we built a linear mixed model, where the log-transformed domestic ancestry rate was introduced as the response variable and the standardized (i.e., center-reduced) recombination rate was included as the explanatory variable (details on the recombination rate estimation can be found in Leitwein et al. [4]). The LGs and 2Mbp windows were incorporated as random effects in the model. We then retrieved the model residuals to highlight genomic regions displaying excess or deficit of domestic ancestry. Heatmaps of the domestic ancestry and the residuals of the model were plotted in R with the package “ggplot2” [45]. In order to obtain unbiased estimates of population genetic diversity, we removed all domestic ancestry tracts previously identified in the ancestry inference step with ELAI [43] from individual genomic data. We quantified the nucleotide diversity based on the population VCF (Variant Call Format) files. Then, we estimated the average nucleotide diversity using the R package PopGenome [46] along sliding windows of 2 Mb weighted by the number of observations (i.e., locus) within each window. The genetic diversity heatmap was plotted in R with the package “ggplot2” [45].

### 2.4. Evaluating the Influence of Genetic Diversity, Stocking Intensity and Recombination Rate on Genomewide Variation in Domestic Ancestry

We examined if and how local genetic diversity, stocking intensity (Table 1) and recombination rate simultaneously affected domestic ancestry at both the genome and linkage group (LG) levels. To estimate the genome-wide recombination rate, we generated a Brook Charr reference genome by anchoring the Brook Charr linkage map from Sutherland et al. [42] to the Artic charr reference genome after controlling for collinearity between those two sister species [31]. Then, MAREYMAP [47] was used to estimate the recombination rate by comparing the physical (pb) and genetic position (cM); the weighted mean recombination rate between the two closest markers was computed for the markers not included in the map [31].

At the genome level, we used linear mixed models where the log-transformed local domestic ancestry rate was included as the response variable. Recombination rate, genetic diversity and stocking intensity were standardized and incorporated as explanatory terms in the models. To consider the non-independency of ancestry estimates across windows, the LGs and the 2-Mb windows were treated as random effects in the models. We used a likelihood ratio test to assess the significance of the tested relationship by comparing the models with and without the explanatory term. We calculated marginal *R*^2^ to quantify the proportion of variance explained by the explanatory variables only.

We then investigated how domestic ancestry was influenced by local genetic diversity, stocking intensity and recombination rate at the linkage group (LG) level. For this, linkage groups were analyzed separately by building linear mixed models which included as fixed effects the best-supported combination of variables (i.e., recombination, genetic diversity and stocking intensity, see Results Section 3.2) evaluated at the genome level. The 2-Mb sliding windows were incorporated as random effect. We evaluated the three possible scenarios presented in Figure 1 across the 42 LG by reporting the slope coefficient of the fixed effects and their 95% confidence intervals. We also calculated *p*-values to evaluate the significance of the three explanatory variables across linkage groups. To correct for multiple testing, we applied the conservative Bonferroni correction and only values with *p* < 0.001 were considered significant (i.e., *α*_corrected_ = 0.05/42 LGs) [48]. All analyses were performed using the R package “LME4” [45].

## 3. Results

### 3.1. Genome-Wide Variation of Domestic Ancestry and Genetic Diversity

The genome-wide level of domestic ancestry rate was estimated for each population using the 405 late-hybrid individuals and an average of 32,442 ± 1799 mapped SNPs per population (Table 1). The domestic ancestry was highly variable among populations with mean rate ranging from 0.0052 (VIE population) to 0.0843 (MIL population) (Table 1). Domestic ancestry was also highly variable among individuals within population, ranging from 0.0000 to 0.6000 (Figure 2). When controlling for recombination rate, we did not observe any shared pattern of deficit or excess of introgression among populations. Consequently, each population presented unique pattern of introgression. Our analyses also showed that some genomic regions showed strong excess of domestic ancestry compared to the rest of the genome and those regions varied among populations. For instance, the GEL population displayed an excess of domestic ancestry for LG 40 and 42 whereas the GRI population showed an ancestry excess in LG 41 and LG 42 (Appendix A).

The level of genetic diversity estimated within 2MB sliding windows was highly variable both among and within populations. The mean population genetic diversity ranged from 0.157 (POR population) to 0.285 (TEM population). The MIL, BRU, GAS and VIE populations also displayed relatively low genetic diversity, with a mean of 0.158, 0.166, 0.168 and 0.176, respectively (Table 1 and Appendix A). Within each population, genetic diversity was also highly variable among individuals ranging from 0.0652 to 0.510 (Table 1 and Appendix A). While some genomic regions displayed particularly high level of genetic diversity, we did not observe any linkage group with a mean diversity significantly lower or higher in comparison to the whole genome (Table 1 and Appendix A).

### 3.2. Influence of Genetic Diversity, Stocking Intensity and Recombination Rate on Domestic Ancestry

At the genome-wide level, we observed more domestic ancestry in low recombining and low-diversity genomic regions (i.e., a pattern of associated overdominance (AOD) Figure 1, scenario A). The global *R*^2^ and the marginal *R*^2^ of the full model (i.e., where the log-transformed domestic ancestry rate was introduced as the response variable and the standardized (i.e., center reduced) recombination rate was included as the explanatory variable) were 0.084 and 0.015, respectively (Figure 3). The likelihood ratio test was significant for genetic diversity, which displayed the strongest effect (*χ*^2^_Diversity_ = 5020.6, *p* < 2.2 × 10^−16^, coefficient slope = −4.95 × 10^−3^ (CI95%: −5.08 × 10^−3^, −4.8× 10^−3^); Figure 3A), stocking intensity (*χ*^2^_Stocking_= 4143.9, *p* < 2.2 × 10^−16^; coefficient slope = 4.4× 10^−3^; (CI95%: 4.3 × 10^−3^, −4.57 × 10^−3^); Figure 3B), as well as recombination rate (*χ*^2^_Recombination_ = 1797.7, *p* < 2.2 × 10^−16^; coefficient slope = −3.41 × 10^−3^ (CI95%: −3.57 × 10^−3^, −3.26 × 10^−3^); Figure 3C). We also found that domestic ancestry was negatively correlated with genetic diversity (slope coefficient *β_Diversity_* = −4.950 × 10^−3^, *p* < 2.2× 10^−16^; Figure 3A) and recombination rate (*β_Recombination_* = −3.418× 10^−3^, *p* < 2.2 × 10^−16^; Figure 3C). The domestic ancestry was positively associated with stocking intensity (*β_Stocking_*= 4.441 × 10^−3^, *p* < 2.2 × 10^−16^) (i.e., Figure 3B).

At the linkage group level, we observed contrasted patterns of relationships between domestic ancestry, local recombination rate, local genetic diversity and stocking intensity among LGs (Figure 4). Ten LGs showed a pattern of associative overdominance with a negative correlation between the domestic ancestry rate and recombination rate and diversity. Among these, seven (LGs 8, 15, 18, 19, 22, 34 and 38; Figure 4) showed a positive correlation with stocking intensity (i.e., Figure 1, scenario A), and three LGs (LGs 9, 11 and 24; Figure 4) showed a higher domestic ancestry for the lakes harboring the lowest stocking intensity. Six LGs (LGs 1, 21, 25, 31, 36 and 39; Figure 4) displayed a pattern of negative epistasis supported by a positive correlation between domestic ancestry rate and recombination rate (Figure 1B,C). Among these, four LGs (LGs 1, 21, 31 and 39; Figure 4) displayed a negative correlation between domestic ancestry rate and genetic diversity (Figure 1, scenario B), and two LGs (LGs 25 and 36) displayed a positive correlation between domestic ancestry rate and all three explanatory variables (Figure 1C). Two LGs did not fit any of the three predicted scenarios (Figure 1), with LG16 displaying same pattern as for negative epistasis associated with the occurrence of beneficial alleles but with more domestic ancestry when the intensity of stocking was lower (Figure 4). LG 17 had more domestic ancestry within high genetic diversity and low recombining genomic regions (Figure 4).

## 4. Discussion

Our study suggests that genetic admixture resulting from supplementation of Brook Charr populations with a domestic strain during the last century produced a lake-specific pattern of ancestry modulated by the interplay between local recombination rate, local genetic diversity and stocking intensity. As previously shown by Létourneau et al. (2018) [28] based on single SNP information, but here based on haplotype information, domestic ancestry increased with stocking intensity. Based on a correlative approach using linear mixed models, our results suggest that associative overdominance (AOD) is the main evolutionary mechanism shaping the genome-wide ancestry landscape among the 22 lakes analyzed. At the linkage group scale, however, our results suggest that negative epistasis effects may also play a role in shaping the local pattern of domestic ancestry, with more domestic ancestry in highly recombining regions. As such, our study suggest that supplementation of small isolated lacustrine populations could result in some form of genetic rescue [15], although potential negative effects possibly reflecting genetic incompatibilities may also occur locally along the genome.

### 4.1. Associative Overdominance as the Main Mechanism Shaping Domestic Ancestry Landscape

Evidence for associative overdominance was supported by prevailing domestic ancestry in low recombining and low diversity genomic regions which could result in the masking of potentially deleterious alleles. AOD effects are particularly expected when the foreign strain used for supplementation harbors a higher level of genetic diversity than populations being supplemented [15,49]. In our study system, the domestic strain does display higher genetic diversity than the wild lacustrine populations [50], which could indeed favor AOD effects. Additionally, populations with small $N_{e}$ such as wild Brook Charr found in small isolated lakes tend to accumulate more deleterious alleles and suffer from inbreeding depression [20,51]. The lacustrine Brook Charr populations studied here are isolated, display small $N_{e}$ [52] and tend to accumulate more deleterious mutations compared to more connected populations [53]. An increasing number of studies has documented that in some circumstances, supplementation of wild populations can result in genetic rescue which translated in an increase of both census and effective population size by mitigating fitness loss associated with inbreeding and reduced genetic polymorphism [20,54,55,56]. To our knowledge, however, only a few studies have provided empirical evidence for the underlying role of associative overdominance [57] and/or the variable consequences of admixture over time following the onset of supplementation [58].

Although AOD could be the main mechanism shaping domestic ancestry patterns at the genome level, the strength of its effect was variable among LGs. The domestic ancestry landscape was dominated by AOD in 10 LGs (Figure 4), where we observed more domestic ancestry in genomic regions characterized by a relatively low recombining rate and low diversity. As both local recombination rate and genetic diversity were variable along the genome, it is not surprising that some linkage groups displayed more AOD signals than others. Additionally, genomic regions tending to accumulate more recessive deleterious mutations will be more subject to AOD effects [59]. To our knowledge, only a few studies have attempted to document genome wide variation of AOD effects. In particular, chromosomal variation of AOD effects has been observed between autosomal and X-chromosomes of *Drosophila melanogaster* [60], whereby autosomal chromosomes maintain higher genetic diversity due to AOD. Variation in AOD effects was also documented in humans, whereby 22 genomic regions displayed unusual peaks of diversity in low recombining regions associated with AOD effects [61].

Overall, while highly significant from a statistical standpoint, the effect size of the correlations between foreign ancestry, recombination, diversity and stocking were small, which can most likely be imputable to the diluting effect of the statistical signals caused by the large amount of data. Given that the effective population sizes (N*e*) of lacustrine Brook Charr populations is generally very small (Median N*e* of 35 CI 32:28 in [53] and 53.6 CI 5.8:1069.4 in [52]), genetic drift is expected to be pronounced. Consequently, it may have contributed to increase the proportion of unexplained genetic variance. Moreover, the relatively recent stocking history (a mean duration of stocking of 17 years over all 22 populations, Appendix A) and the time at maturity of three years for the Brook Charr [31,37] likely decreased our ability to detect a strong genomic signature of AOD and negative epistasis. Indeed, it has been shown with simulated data that the ability to detect genetic signals of adaptive and maladaptive introgression increases with the number of generations after the introduction of the foreign genotypes [23].

### 4.2. Negative Epistasis also Contributes to the Dynamics of Domestic Ancestry Landscape

Negative epistasis also interplayed with AOD in modulating the dynamics of the genome-wide variation in domestic ancestry rate, with more domestic ancestry observed in high recombining regions of six LGs. In theory, hybridization between two species and even divergent populations of the same species may lead to “Dobzhansky–Muller” incompatibilities [16,19,62,63,64]. In particular, individuals bred in captivity may adapt to captive conditions and can then develop genetic incompatibilities with their wild counterparts, leading to outbreeding depression when they are released into the wild [65,66], in addition to potentially diluting local adaptation. Indeed, outbreeding depression has previously been documented in wild populations following supplementation with captive populations [67], and these cases have received considerable attention from conservation biologists over the last three decades [20,68,69,70,71,72]. Here, it is therefore not surprising to detect in some genomic regions the molecular signature of negative epistasis since the domestic strain has been bred in captivity for more than 100 years (or at least 30 generations). Along with signals of negative epistasic effects, we were also expecting to observe a higher level of domestic ancestry in genomic regions harboring the highest level of genetic diversity, given that regions with high recombination rate generally tend to show high genetic diversity [73]. However, and contrary to expectations, we observed more domestic ancestry in low diversity genomic regions for four LGs. This pattern could hypothetically be explained by the presence of beneficial domestic alleles as in the case of fixation of a beneficial allele through selective sweeps, which may result in locally reduced neutral variation around such beneficial alleles [74].

### 4.3. Perspectives for Conservation

Assessing the evolutionary mechanisms at play and their consequences in the context of population supplementation can help to better understand the tradeoff between costs and benefits of human-driven hybridization that may result from supplementation practices [13]. Stocking programs in a fishery context are conducted to increase the census size of fish populations and the number of fish that can be harvested but often neglect the evolutionary consequences of introducing foreign genotypes in the supplemented populations. Ultimately, being able to more accurately predict whether the introduction of foreign alleles will improve (i.e., genetic rescue) or decrease (i.e., outbreeding depression) the mean fitness of the supplemented populations could inform management decisions and assist in designing conservation programs [20,58,67,69]. Here, however, our results suggest that the outcomes of supplementation are largely unpredictable as they showed that each population displayed a unique pattern of ancestry landscape following stocking operations using the same domestic strain. After controlling for local variation in recombination rate along the genome, our analyses provided little evidence for shared patterns of domestic ancestry among populations. Such population-specific ancestry patterns could be partly explained by the highly variable pattern of genome-wide genetic diversity among populations, a likely consequence of pronounced genetic drift being enhanced by small effective population sizes of these populations [52] and very limited gene flow between them [53]. Furthermore, local adaptation to contrasted environmental conditions prevailing in the different lakes [53] likely modulates the variation in genetic diversity along the genome via soft selective sweeps. This may favor or disfavor the foreign alleles introduced and could therefore contribute to generate the population-specific ancestry landscape observed in our study and in previous ones [32,50]. Admittedly, however, in the absence of phenotypic and/or fitness information predating and postdating supplementation, the resulting demographic consequences of the opposite effects of AOD and negative epistasis we have documented here remain hypothetical and deserve further investigation. Nevertheless, our results suggest that supplementation could possibly be acting as a form of genetic rescue in those small isolated populations characterized by very small N*e* and a low level of genetic diversity. Yet, this potentially positive asset must be considered cautiously as the use of the same domestic strain to supplement multiple populations can lead to an homogenization of the population genetic structure [50] and to the dilution of local adaptation via the disruption of co-adapted gene networks [32,75]. Moreover, our results also suggest that, alongside the main positive effects of AOD, negative epistasis may also determine local variation of domestic ancestry, which may in turn be at the source of outbreeding depression, ultimately impacting the outcomes of supplementations [76,77]. Finally, while AOD effects decrease with time after hybridization, maladaptive effects could arise later on, especially in populations with small *Ne* [78].

## 5. Conclusions

To conclude, our study emphasizes the need for conservation biologists to examine how the complex interplay between the introduction pressure of exogenic genetic makeup, genetic diversity and recombination rate along the genome modulates introgression landscapes of recipient populations. As such, this study offers new avenues for the study of the mechanisms shaping the evolution of exploited populations and regulating the dynamics of hybridization, either occurring naturally (e.g., hybrid zones) or caused by humans.

## Figures and Tables

**Figure 1 genes-12-00524-f001:**
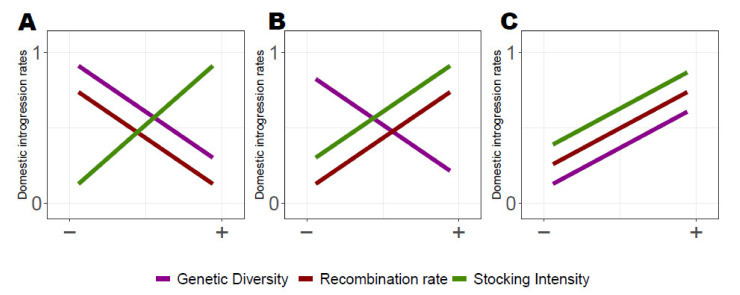
Predicted relationships between domestic ancestry rate, local recombination rate, local genetic diversity and stocking intensity. (**A**) Predicted relationship expected with associative overdominance (AOD), where more domestic ancestry are observed in low-diversity genomic region by masking the effects of potentially recessive deleterious mutation [13]. (**B**,**C**) Predicted relationship with negative epistasis where loci of incompatibility results in lower domestic ancestry in low recombining genomic regions. (**B**) Predicted relationship in presence of potential local beneficial alleles where lower diversity is expected due to the fixation of the beneficial alleles and higher introgression of domestic ancestry. (**C**) Predicted higher domestic ancestry in low-diversity genomic regions, namely due to the purge of incompatibility alleles [13,34]. X-axis illustrates the range of variation for each of the three variables (i.e., from lower to higher).

**Figure 2 genes-12-00524-f002:**
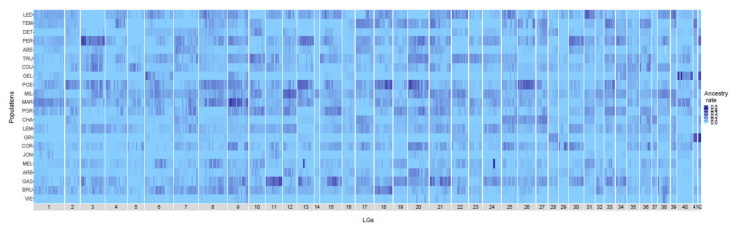
Heatmap of the proportion of domestic ancestry rate along the 42 Brook Charr linkage groups (LGs) for each lake. Lakes are ordered as a function of stocking intensity (less stocked lakes at the bottom). Labels are detailed in Table 1.

**Figure 3 genes-12-00524-f003:**
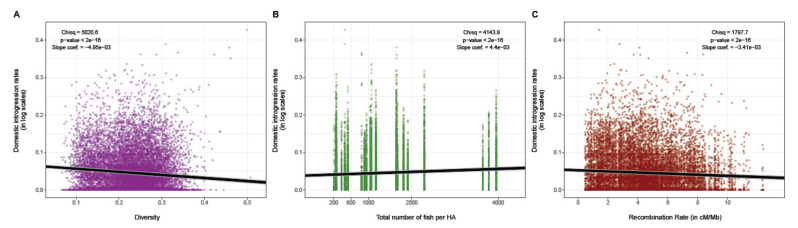
Whole genome relationship between the domestic ancestry and (**A**) genetic diversity, (**B**) stocking intensity and (**C**) local recombination rate, assessed with a generalized linear mixed model (GLM; *R*^2^c = 0.084, *R*^2^m = 0.015, AIC = −1952637, χ^2^ = 4143.9, *p* < 2.2× 10^−16^ (**B**); *β_Recombination_* = −3.418 × 10^−3^, *β_Diversity_* = −4.950 × 10^−3^, *β_Stocking_* = 4.441 × 10^−3^, *p* < 2.2× 10^−16^). Grey lines represent the 95% confidence interval.

**Figure 4 genes-12-00524-f004:**
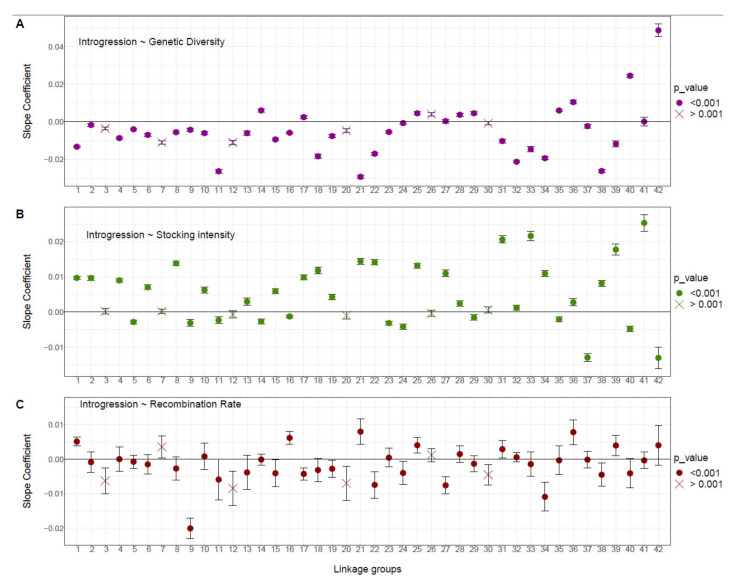
Relationship between the domestic ancestry rate and (**A**) genetic diversity, (**B**) stocking intensity and (**C**) local recombination rate for each linkage group. Slope coefficients of the GLM model are reported for each 42 Brook Charr linkage group. Dots represent the significance of each model with circles and crosses representing *p* < 0.001 and *p* > 0.001, along with the 95% confidence interval, respectively.

**Table 1 genes-12-00524-t001:** Description of the 22 sampled Brook Charr lakes in Québec.

Reserve	Lake	Label	N_samples_filters	N_late_hybrids	N_SNPs_mapped	lakes_size	total_ha	Mean Ancestry Rate	Mean Genetic Diversity
Mastigouche	Abénakis	ABE	21	16	31529	5	1915	0.0253	0.259
	Arbout	ARB	28	24	31713	5	380	0.0261	0.259
	Chamberlain	CHA	20	19	33290	18	958	0.0324	0.261
	Cougouar	COU	29	28	33455	8	1669	0.0426	0.252
	Deux-Etapes	DET	28	22	33580	12.3	3647	0.0282	0.246
	Gélinotte	GEL	25	18	31377	5	1652	0.0209	0.271
	Grignon	GRI	23	17	33298	29.6	846	0.0186	0.26
	Jones	JON	26	17	30553	28	468	0.0062	0.25
	Ledoux	LED	26	12	30941	13.7	3956	0.0663	0.236
	Lemay	LEM	28	19	30253	19.1	914	0.0538	0.23
Saint-Maurice	Brulȏt	BRU	25	17	35463	8.1	247	0.0557	0.166
	Corbeil	COR	23	16	28071	9.5	526	0.0491	0.261
	Gaspard	GAS	28	14	34844	11.6	259	0.0766	0.168
	Maringouins	MAR	26	17	31102	6.2	1073	0.0734	0.23
	Melchior	MEL	26	20	30720	4.4	455	0.038	0.212
	Milord	MIL	25	16	41035	46.7	1175	0.0843	0.158
	Perdu	PER	24	17	32484	22.1	2293	0.075	0.247
	Porc-Epic	POE	26	17	32100	2.7	1648	0.0829	0.247
	Portage	POR	27	12	34415	46.9	1044	0.0615	0.157
	Tempȇte	TEM	17	11	32852	12.5	3784	0.046	0.286
	À la truite	TRU	26	21	31741	6.5	1814	0.0597	0.26
	Vierge	VIE	26	12	33747	5.7	208	0.0052	0.176

N_samples_filters: Number of individuals after filtering; N_late_hybrids: Number of individuals identified as late hybrids; N_SNPs_mapped: Number of mapped SNPs; lakes_size: Laked size in ha; total_ha: The total number of fish stocked/ha; Mean ancestry rate: Mean ancestry rate (estimated from ELAI) per lakes; Mean genetic diversity: Mean genetic diversity based on SNPs within 2 Mb sliding windows.

## Data Availability

The data presented in this study are openly available in Létourneau et al. [3] at Dryad Digital Repository: https://doi.org/10.5061/dryad.s5qt3.

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
