# Peer review of "Associative Overdominance and Negative Epistasis Shape Genome-Wide Ancestry Landscape in Supplemented Fish Populations"

_genes, 2021, doi:10.3390/genes12040524_

Round 1

Reviewer 1 Report

The proposed manuscript (ms) greatly contributes to the study of population genetics, especially how is wild population of fish species influenced by gene introgression from another (domestic) population. I have not found any fundamental issues in proposed ms. The ms is technically sound, presented in an intelligible fashion and written in good-quality English. Appropriately chosen terms and abbreviations are clearly explained and used for better understanding. Impact of the study is well highlighted. Authors follow up on their previous research (Leitwein et al. 2019) which points out on their great ability to publish statistical/genomic data.

I Have only a few minor suggestions:

1) In introduction, there is stated that biological invasion is one of the major causes of genetic admixture within wild and non-local populations. There are another relevant evidences of genetic introgression in fish population as a result of human-mediated translocation. Carassius gibelio is introduced and highly invasive species and C. carassius native and threatened. Mitochondrial and nuclear markers, and fluorescence in situ hybridization painting revealed genome admixture within aforementioned species (Knytl et al. 2013, 2018). This introgression greatly benefits non-native Carassius in the term of successful colonization of new areas. Could authors enrich the ms introduction by at least one of Knytl et al. studies?

2) In the first paragraph of the page 3, there is "(i.e. among the genome)". Should not be "genome" in plural?

3) Figure 4 legend: double check "rate" in the first sentence.

4) Population size label (Ne), sometimes italics is used sometimes does not (Ne x Ne).

References:

Knytl M, Kalous L, Symonová R, Rylková K, Ráb P. Chromosome studies of European cyprinid fishes: cross-species painting reveals natural allotetraploid origin of a Carassius female with 206 chromosomes. Cytogenet Genome Res. 2013;139(4):276-83. doi: 10.1159/000350689. Epub 2013 May 4. PMID: 23652770.

Knytl M, Kalous L, Rylková K, Choleva L, Merilä J, Ráb P. Morphologically indistinguishable hybrid Carassius female with 156 chromosomes: A threat for the threatened crucian carp, C. carassius, L. PLoS One. 2018 Jan 23;13(1):e0190924. doi: 10.1371/journal.pone.0190924. PMID: 29360831; PMCID: PMC5779652.

Leitwein M, Cayuela H, Ferchaud AL, Normandeau É, Gagnaire PA, Bernatchez L. The role of recombination on genome-wide patterns of local ancestry exemplified by supplemented brook charr populations. Mol Ecol. 2019 Nov;28(21):4755-4769. doi: 10.1111/mec.15256. Epub 2019 Oct 22. PMID: 31579957.

Author Response

Reviewer 1

The proposed manuscript (ms) greatly contributes to the study of population genetics, especially how is wild population of fish species influenced by gene introgression from another (domestic) population. I have not found any fundamental issues in proposed ms. The ms is technically sound, presented in an intelligible fashion and written in good-quality English. Appropriately chosen terms and abbreviations are clearly explained and used for better understanding. Impact of the study is well highlighted. Authors follow up on their previous research (Leitwein et al. 2019) which points out on their great ability to publish statistical/genomic data.

>We thank the reviewer for acknowledging our work

I Have only a few minor suggestions:

  • In introduction, there is stated that biological invasion is one of the major causes of genetic admixture within wild and non-local populations. There are another relevant evidences of genetic introgression in fish population as a result of human-mediated translocation. Carassius gibelio is introduced and highly invasive species and C. carassius native and threatened. Mitochondrial and nuclear markers, and fluorescence in situ hybridization painting revealed genome admixture within aforementioned species (Knytl et al. 2013, 2018). This introgression greatly benefits non-native Carassius in the term of successful colonization of new areas. Could authors enrich the ms introduction by at least one of Knytl et al. studies?

>Thank you for the suggestions. Both citations have been added to the introduction

2) In the first paragraph of the page 3, there is "(i.e. among the genome)". Should not be "genome" in plural?

>Indeed we have now put genome in plural

3) Figure 4 legend: double check "rate" in the first sentence.

>This has been corrected

4) Population size label (Ne), sometimes italics is used sometimes does not (Ne x Ne).

>This has been corrected

References:

Knytl M, Kalous L, Symonová R, Rylková K, Ráb P. Chromosome studies of European cyprinid fishes: cross-species painting reveals natural allotetraploid origin of a Carassius female with 206 chromosomes. Cytogenet Genome Res. 2013;139(4):276-83. doi: 10.1159/000350689. Epub 2013 May 4. PMID: 23652770.

Knytl M, Kalous L, Rylková K, Choleva L, Merilä J, Ráb P. Morphologically indistinguishable hybrid Carassius female with 156 chromosomes: A threat for the threatened crucian carp, C. carassius, L. PLoS One. 2018 Jan 23;13(1):e0190924. doi: 10.1371/journal.pone.0190924. PMID: 29360831; PMCID: PMC5779652.

Leitwein M, Cayuela H, Ferchaud AL, Normandeau É, Gagnaire PA, Bernatchez L. The role of recombination on genome-wide patterns of local ancestry exemplified by supplemented brook charr populations. Mol Ecol. 2019 Nov;28(21):4755-4769. doi: 10.1111/mec.15256. Epub 2019 Oct 22. PMID: 31579957.

Reviewer 2 Report

I have read the manuscript entitled “Associative-overdominance and negative epistasis shape genome-wide ancestry landscape in supplemented fish populations” by Leitwein et al. This work examined 522 Brook Charr from 22 small lake populations genotyped at ~32,400 mapped SNPs and 37 individuals from the domestic brood stock. Each lake showed a pattern of an ancestral population that was supplemented with domestic strains. The goal of this study was to test whether associative overdominance or negative epistasis appeared stronger in shaping the domestic introgression rates between native and domestic brook trout. The study has a solid set of testable hypothesis, and the study design tests these hypotheses. It is a clever experiment in a complicated topic with complicated data. Ten linkage groups showed a pattern of associative overdominance and six showed a pattern of negative epistasis.

While the paper is generally well-written, there are many cases in which subjects and verbs are not in the same tense. Please check for subject-verb agreement throughout. Also, from a critical standpoint, what are the chances that the same results could have been achieved by chance alone? There are 42 linkage groups so the finding of 10 showing AOD and 6 with negative epistasis are not overwhelming. It would strengthen the paper to address this point, and the paper could be criticized if the trends could be achieved by chance alone. I also have some concern about the significance shown in Figure 3A that I describe below. Finally, slight reorganization of the discussion would be helpful. Consider a single paragraph in the discussion that considers the shortfalls of the study, and each subtopic focusing on a conclusion.

Abstract:

I found the abstract more difficult to read prior to reading the manuscript because several terms that are likely unfamiliar to most readers are undefined. The abstract will be much more clear if you define: “foreign ancestry”, and “domestic strain”, and clarify which strain is native to the 22 lakes and which is introduced. Use these terms throughout the paper.

  1. Not sure what ‘foreign ancestry’ is.
  2. ‘In a system of supplemented lacustrine populations with a single domestic strain,’ Does this mean that there was initially one strain in the lakes or does it mean it was supplemented with a single strain?
  3. Does “ from 22 small lacustrine populations” mean you looked at 22 lakes?
  4. Suggest “highly variable patterns
  5. It is unclear what you mean by domestic vs. foreign ancestry (please define in general terms such as introduced or native).
  6. Suggest “lakes specific ancestry patterns were mainly due”
  7. Suggest “ (AOD) effects among populations
  8. What kind of “apparent positive effects” are you referring to?
  9. You refer also to “recipient” and “foreign” population – this is good – but overall a definition of these as well as “domestic” would be helpful.
  10. Suggest “precursory signs”.
  11. Consider rewording last sentence of abstract; “it’s” appears informal in this context.
  12. Despite the title, the Abstract does not mention the finding of negative epistasis.

Introduction:

  1. This is a large suite of potential predictors. “We predicted three alternative evolutionary scenarios depending on the interplay between genome-wide domestic ancestry rate variation, local recombination rate, local genetic diversity, and lake-specific stocking history”
  2. I think referring to Figure 1 is an excellent way to describe your hypotheses. However, I cannot follow Figure 1 because I do not understand the x-axes. I like the color coding of genetic diversity, recombination rate, and stocking intensity throughout the paper.
  3. Overall, the introduction does a good job introducing terms and explaining the experimental design, with the exception of the last paragraph, which is quite complicated. For the last paragraph of the introduction, a few sentences describing in words your methodology would be helpful. For example, “We considered the relationships between genetic diversity and domestic introgression rate, between stocking intensity and domestic introgression rate, and between recombination rate and domestic introgression rate when drawing conclusions about the effects of AOD and negative epistasis.

Methods:

  1. In this sentence “To do so, we built a linear mixed model where the log-transformed domestic ancestry rate was introduced as the response variable and the standardized (i.e., center-reduced) recombination rate was included as the explanatory variable.” How did you calculate recombination rate? You mention it in the next section – please describe at first mention.
  2. How did you decide that 2Mb was a good size for your sliding windows? And what was the step size between sliding windows?

Results:

  1. I am confused by this sentence: “When controlling for recombination rate, we did not observe any shared pattern of deficit or excess of introgression among populations which were in fact unique to each of them.” Is that another way of saying that introgression did not differ among lake populations?
  2. Consider “The level of genetic diversity estimated within 2MB sliding windows was”
  3. This sentence is drawing conclusions from your results “At the genome-wide level, our results revealed a predominant pattern of associated overdominance (AOD) whereby more domestic ancestry was observed in low recombin-ing and low-diversity genomic regions (Figure. 1, scenario A).”; therefore it should be in your discussion section.
  4. Please reiterate what the full model is in this sentence: “The global and the mar-ginal of the full model were 0.084 and 0.015, respectively”
  5. In this sentence: “The likelihood ratio test was significant for recombination rate (χ…” please describe figure in order (e.g A, B, C).
  6. Regarding this sentence: “As expected, the domestic ancestry was positively associated with stocking in “, the results sectionshould simply state the findings, without bias or interpretation.
  7. Please add a bit more explanation in this section: “Two LGs did not fit the three predicted scenarios (Figure 1), with LG16 displaying a pattern of negative epistasis associated with the occurrence of beneficial alleles but with more domestic ancestry when the intensity of stocking was lower (Figure 4). LG 17 had more domestic ancestry within high genetic diversity and low recombining genomic regions (Figure 4).” How does LG16 show a pattern of negative epistasis when it does not fit one of the 3 scenarios that were designed to test for negative epistasis?

Discussion

  1. It seems this sentence would be better to start with ‘However’: “Moreover, populations with small ?? _such as wild Brook Charr found in small isolated lakes tend to accumulate more deleterious alleles and suffer from”
  2. Check subject-verb agreement “To our knowledge however, only a few study has provided empirical…”
  3. What is IC here ? “Median Ne of 35 IC 32:28 ”
  4. Consider “predict” rather than “predicting” “Ultimately, being able to more accurately predicting”

Figure 1. Please define X-axis. Is it A. x-axis is AOD, B. x-axis is ? This is a key figure – please describe it so clearly that even a non-geneticist could understand it.

Figure 2. This figure would benefit from much larger legend and axis text. I cannot see the legend clearly.

Figure 3. Please increase font size throughout this figure. Is it possible your p-values are overinflated by the number of SNPs? In each of these images, the trends do not look strong but the p-values are very low.

This is a very informative figure. I am somewhat concerned that the pattern observed in panel A is real. There are several outliers but very little data when genetic diversity is high (>0.4). Is it possible that these outliers are driving the trend? Consider running the regression again but only using regions with a lot of data (0.1>diversity >3.5).

Table 1. How is ancestry rate calculated? Please note the source for this calculation.

Author Response

Revier2

I have read the manuscript entitled “Associative-overdominance and negative epistasis shape genome-wide ancestry landscape in supplemented fish populations” by Leitwein et al. This work examined 522 Brook Charr from 22 small lake populations genotyped at ~32,400 mapped SNPs and 37 individuals from the domestic brood stock. Each lake showed a pattern of an ancestral population that was supplemented with domestic strains. The goal of this study was to test whether associative overdominance or negative epistasis appeared stronger in shaping the domestic introgression rates between native and domestic brook trout. The study has a solid set of testable hypothesis, and the study design tests these hypotheses. It is a clever experiment in a complicated topic with complicated data. Ten linkage groups showed a pattern of associative overdominance and six showed a pattern of negative epistasis.

While the paper is generally well-written, there are many cases in which subjects and verbs are not in the same tense. Please check for subject-verb agreement throughout. Also, from a critical standpoint, what are the chances that the same results could have been achieved by chance alone? There are 42 linkage groups so the finding of 10 showing AOD and 6 with negative epistasis are not overwhelming. It would strengthen the paper to address this point, and the paper could be criticized if the trends could be achieved by chance alone. I also have some concern about the significance shown in Figure 3A that I describe below. Finally, slight reorganization of the discussion would be helpful. Consider a single paragraph in the discussion that considers the shortfalls of the study, and each subtopic focusing on a conclusion.

 >We thank the reviewer for these suggestions. English has been checked. Please see below our point by point answers. Concerning the 42 linkage groups, we understand the reviewer points, however we have been very stringent in our analysis as conservative false discovery rate have been applied for each LGs models (i.e. Bonferroni false discovery rate). Added to our conservative approaches, we also did the analysis at the global scale (all LGs together) and found that AOD was the main mechanism. All together we are pretty confident that our results are not only by chances.

Abstract:

I found the abstract more difficult to read prior to reading the manuscript because several terms that are likely unfamiliar to most readers are undefined. The abstract will be much more clear if you define: “foreign ancestry”, and “domestic strain”, and clarify which strain is native to the 22 lakes and which is introduced. Use these terms throughout the paper.

  1. Not sure what ‘foreign ancestry’ is.

> “foreign” has been removed here for more clarity as it was not needed

  1. ‘In a system of supplemented lacustrine populations with a single domestic strain,’ Does this mean that there was initially one strain in the lakes or does it mean it was supplemented with a single strain?

>We meant supplemented with a single strain

We rephrased as follows: “In a system of lacustrine populations supplemented with a single domestic strain”.

  1. Does “ from 22 small lacustrine populations” mean you looked at 22 lakes?

>Yes that what we meant

  1. Suggest “highly variable patterns

>This has been corrected

  1. It is unclear what you mean by domestic vs. foreign ancestry (please define in general terms such as introduced or native).

>the domestic strains meant a strain rose in hatchery and introduced in natural population we specified as follow: “supplementation with domestic strains (i.e. human-introduced strains).”

  1. Suggest “lakes specific ancestry patterns were mainly due”

>This has been corrected

  1. Suggest “ (AOD) effects among populations “

>This has been corrected

  1. What kind of “apparent positive effects” are you referring to?

>we rephrased as follows “(i.e potential positive effects due to the masking of possible deleterious alleles in low recombining region)”

  1. You refer also to “recipient” and “foreign” population – this is good – but overall a definition of these as well as “domestic” would be helpful.

>we replaced recipient and foreign by native and introduced for more consistency in the abstract

  1. Suggest “precursory signs”.

>This has been corrected

  1. Consider rewording last sentence of abstract; “it’s” appears informal in this context.

>We rephrased as follows “Consequently, in order to improve conservation practices and management strategies, it became necessary  to assess the consequences of supplementation at a the population level by taking into account both genetic diversity and stocking intensity when available.”

  1. Despite the title, the Abstract does not mention the finding of negative epistasis.

>Thank you. We now added “Local negative effects such as negative epistasis (i.e. potential genetic incompatibilities between the native and the introduced population) potentially reflecting precursory signs of outbreeding depression were also observed at a chromosomal scale.”

Introduction:

  1. This is a large suite of potential predictors. “We predicted three alternative evolutionary scenarios depending on the interplay between genome-wide domestic ancestry rate variation, local recombination rate, local genetic diversity, and lake-specific stocking history”

>We rephrased as follows: “We predicted three alternative evolutionary scenarios considering the relationship between genome-wide domestic ancestry rate variation  and (i)local recombination rate (ii)local genetic diversity and (iii) lake-specific stocking history (Figure 1 A-C).”

  1. I think referring to Figure 1 is an excellent way to describe your hypotheses. However, I cannot follow Figure 1 because I do not understand the x-axes. I like the color coding of genetic diversity, recombination rate, and stocking intensity throughout the paper.

>On Figure 1 the (-) and (+) represent the intensity of the 3 variables  “less” or “more” for each variable e.g. 0 to 12 for the recombination rate, 0 to 0.5 for the diversity and 200 to 4000 for the stocking intensity. Those ranges of values are simply meant to illustrate the scale of variation in relative terms.  

This has been added to Figure 1 legends “X-axis illustrates the range of variation for each of the three variables (i.e. from lower to higher).”

  1. Overall, the introduction does a good job introducing terms and explaining the experimental design, with the exception of the last paragraph, which is quite complicated. For the last paragraph of the introduction, a few sentences describing in words your methodology would be helpful. For example, “We considered the relationships between genetic diversity and domestic introgression rate, between stocking intensity and domestic introgression rate, and between recombination rate and domestic introgression rate when drawing conclusions about the effects of AOD and negative epistasis.

>We rephrased as follows: “We predicted three alternative evolutionary scenarios considering the relationships between genome-wide domestic ancestry rate variation  and (i)local recombination rate (ii)local genetic diversity and (iii) lake-specific stocking history (Figure 1 A-C). “

And at the end “Then, we determined which one of the aforementioned alternative scenarios (i.e AOD vs negative epistasis) best explained the observed patterns of domestic ancestry across linkage groups.

Methods:

  1. In this sentence “To do so, we built a linear mixed model where the log-transformed domestic ancestry rate was introduced as the response variable and the standardized (i.e., center-reduced) recombination rate was included as the explanatory variable.” How did you calculate recombination rate? You mention it in the next section – please describe at first mention.

>This has been corrected

  1. How did you decide that 2Mb was a good size for your sliding windows? And what was the step size between sliding windows?

>We used 2Mb for consistency with the previous study, each window are following each other, there is no overlapping. 

Results:

  1. I am confused by this sentence: “When controlling for recombination rate, we did not observe any shared pattern of deficit or excess of introgression among populations which were in fact unique to each of them.” Is that another way of saying that introgression did not differ among lake populations?

>Here we meant that each pattern of introgression is different between each population and there is no similar pattern (i.e. positive or negative introgression shared by all lakes).

We rephrased as follows: “When controlling for recombination rate, we did not observe any shared pattern of deficit or excess of introgression among populations. Consequently, each population present unique pattern of introgression. “

  1. Consider “The level of genetic diversity estimated within 2MB sliding windows was”

> Corrected

  1. This sentence is drawing conclusions from your results “At the genome-wide level, our results revealed a predominant pattern of associated overdominance (AOD) whereby more domestic ancestry was observed in low recombin-ing and low-diversity genomic regions (Figure. 1, scenario A).”; therefore it should be in your discussion section.

>We rephrased as follows “At the genome-wide level, we observed more domestic ancestry in low recombining and low-diversity genomic regions (i.e. pattern of associated overdominance (AOD) Figure. 1, scenario A).”

  1. Please reiterate what the full model is in this sentence: “The global R² and the mar-ginal R² of the full model were 0.084 and 0.015, respectively”

>We rephrased as follows ‘The global and the marginal of the full model (i.e. where the log-transformed domestic ancestry rate was introduced as the response variable and the standardized (i.e., center-reduced) recombination rate was included as the explanatory variable)”

  1. In this sentence: “The likelihood ratio test was significant for recombination rate (χ…” please describe figure in order (e.g A, B, C).

> This has been corrected as follows “The likelihood ratio test was significant for genetic diversity, which displayed the strongest effect (χ²Diversity= 5020.6, p < 2.2e−16, coefficient slope =-4.95e-03 [IC95%: -5.08e-03, -4.8e-03]; Figure 3A),  stocking intensity (χ²Stocking= 4143.9, p < 2.2e−16; coefficient slope= 4.4e-03; [IC95%: 4.3e-03, -4.57e-03]; Figure 3B), as well as recombination rate (χ²Recombination= 1797.7, p < 2.2e−16; coefficient slope = -3.41e-03 [IC95%: -3.57e-03, -3.26e-03]; Figure 3C). We also found that domestic ancestry was negatively correlated with genetic diversity (slope coefficient βDiversity= -4.950e-03, p < 2.2e-16; Figure 3A) and recombination rate (βRecombination= -3.418e-03, p < 2.2e-16; Figure 3C). As expected, the domestic ancestry was positively associated with stocking intensity (βStocking= 4.441e-03, p < 2.2e-16) (i.e. Figure 3B).”

  1. Regarding this sentence: “As expected, the domestic ancestry was positively associated with stocking in “, the results sectionshould simply state the findings, without bias or interpretation.

>”As expected” has been removed

  1. Please add a bit more explanation in this section: “Two LGs did not fit the three predicted scenarios (Figure 1), with LG16 displaying a pattern of negative epistasis associated with the occurrence of beneficial alleles but with more domestic ancestry when the intensity of stocking was lower (Figure 4). LG 17 had more domestic ancestry within high genetic diversity and low recombining genomic regions (Figure 4).” How does LG16 show a pattern of negative epistasis when it does not fit one of the 3 scenarios that were designed to test for negative epistasis?

>We rephrased as follows: “Two LGs did not fit any of the three predicted scenarios (Figure 1), with LG16 displaying same pattern as for negative epistasis associated with the occurrence of beneficial alleles but with more domestic ancestry when the intensity of stocking was lower (Figure 4).”

Discussion

  1. It seems this sentence would be better to start with ‘However’: “Moreover, populations with small ?? _such as wild Brook Charr found in small isolated lakes tend to accumulate more deleterious alleles and suffer from”

>For this statement we meant “additionally, populations with...” This has now been corrected.  

  1. Check subject-verb agreement “To our knowledge however, only a few study has provided empirical…”

>we replaced has by have in the sentence

  1. What is IC here ? “Median Ne of 35 IC 32:28 ”

>we meant CI for Confidence Interval, this has been corrected

  1. Consider “predict” rather than “predicting” “Ultimately, being able to more accurately predicting”

>”predicting” has been replace by “predict”

Figure 1. Please define X-axis. Is it A. x-axis is AOD, B. x-axis is ? This is a key figure – please describe it so clearly that even a non-geneticist could understand it.

 >The X-axis are simply meant to illustrate the scale of variation in relative terms for each variables (from lower to higher).  

Figure 2. This figure would benefit from much larger legend and axis text. I cannot see the legend clearly.

>We increased as much as possible legends and axes 

Figure 3. Please increase font size throughout this figure. Is it possible your p-values are overinflated by the number of SNPs? In each of these images, the trends do not look strong but the p-values are very low.

>The font size has been increased. We didn’t expect strong effect as they are small effects along the genomes.

This is a very informative figure. I am somewhat concerned that the pattern observed in panel A is real. There are several outliers but very little data when genetic diversity is high (>0.4). Is it possible that these outliers are driving the trend? Consider running the regression again but only using regions with a lot of data (0.1>diversity >3.5).

>  We introduced the 2Mb windows in order to avoid artifacts driven by outliers. Moreover, there is so many data that potential outliers do not impact the models.

Table 1. How is ancestry rate calculated? Please note the source for this calculation.

>This has been added to Table 1 legend